# Evaluating Food Packaging Waste in Schools: A Systematic Literature Review

**DOI:** 10.3390/ijerph19095607

**Published:** 2022-05-05

**Authors:** Jessica Heiges, Danielle L. Lee, Laura Vollmer, Kate Wobbekind, Hannah R. Thompson, Wendi Gosliner, Kristine A. Madsen, Kate O’Neill, Lorrene D. Ritchie

**Affiliations:** 1Department of Environmental Science, Policy, and Management, University of California Berkeley, Berkeley, CA 94720, USA; kmoneill@berkeley.edu; 2Nutrition Policy Institute, Division of Agriculture and Natural Resources, University of California, Oakland, CA 94607, USA; wgosliner@ucanr.edu (W.G.); lritchie@ucanr.edu (L.D.R.); 3Division of Agriculture and Natural Resources, University of California Cooperative Extension, Half Moon Bay, CA 94019, USA; lvollmer@ucanr.edu; 4Student Nutrition Services, San Francisco Unified School District, San Francisco, CA 94102, USA; wobbekindk@sfusd.edu; 5School of Public Health, University of California Berkeley, Berkeley, CA 94720, USA; thompsonh@berkeley.edu (H.R.T.); madsenk@berkeley.edu (K.A.M.)

**Keywords:** municipal solid waste, school foodservice, food packaging waste, waste audits, municipal solid waste methodologies, data collection instrument

## Abstract

Public schools in the U.S. generate about 14,500 tons of municipal solid waste daily, and approximately 42% of that is food packaging generated by school foodservice, contributing significantly to the global packaging waste crisis. This literature review summarizes methods used to evaluate food packaging waste in school foodservice. This review has two objectives: first, to understand which methodologies currently exist to evaluate food packaging waste generation and disposal in school foodservice; and second, to describe the creation of and share a practical standardized instrument to evaluate food packaging waste generation and disposal in school foodservice. A systematic review was conducted using the following search terms: solid waste, school, cafeteria and food packaging, waste, and school. The final review included 24 studies conducted in school environments (kindergarten through twelfth grade or college/university), 16 of which took place in the U.S. Food packaging waste evaluations included objective methods of waste audits, models, and secondary data as well as subjective methods of qualitative observations, questionnaires, interviews, and focus groups. Large variation exists in the settings, participants, designs, and methodologies for evaluating school foodservice packaging waste. Lack of standardization was observed even within each methodology (e.g., waste audit). A new instrument is proposed to support comprehensive and replicable data collection, to further the understanding of school foodservice food packaging waste in the U.S., and to reduce environmental harms.

## 1. Introduction

Municipal solid waste (MSW) is the landfill, compost, and recycling waste generated after a product is produced and leaves the retailer or distributor, such as food scraps, household cleaning containers, textiles, and food packaging material. Over 2.2 billion metric tons of MSW were generated globally in 2018 and this waste is predicted to increase to almost 3.5 billion metric tons by 2050 [1]. MSW has a direct impact on human and planetary health and poses increasing challenges to ensure it is managed in a way that minimizes harms, optimizes resource recovery, and mitigates climate change [2]. High income countries are the largest MSW generators. For example, in 2018 the United States generated an estimated nearly 4.9 pounds (2.22 kg) of MSW per capita each day [3] compared to lower income countries, such as those in the East Asia and Pacific region, which generated an estimated 1.2 pounds (0.56 kg) of MSW per capita each day [1].

Public schools in high income countries are a major source of MSW. There are roughly 50 million students that attend public schools (prekindergarten through twelfth grade) in the U.S. [4], a setting that is collectively responsible for 14,500 tons of MSW daily [5]. Nearly half (~42%) of MSW from the U.S. school setting is food packaging waste generated by school foodservice (e.g., Styrofoam trays, milk cartons) [6]. A 2016 study in upstate New York found a high tonnage rate of food packaging waste from school foodservice, despite the bulk of the materials being lightweight, demonstrating the high usage of food packaging materials [7].

All along the food packaging supply chain, there are harms to environmental and human health that disproportionately impact economically disadvantaged communities and communities of color [8,9]; communities that are more likely to participate in school foodservice programs [10]. The supply chain harms to human health stem from four major sources: greater exposure to and larger consumption of toxic materials due to the types of foods and food packaging made available in these communities [11]; living in communities within 3 miles (high-risk) or up to 25 miles (vulnerable) of a hazardous industrial or commercial facility associated with the food packaging supply chain and waste disposal [12,13]; working in high-risk facilities with large toxic chemical exposure such as the refinery and manufacturing plants that create plastic and disposable food packaging, and the MSW disposal facilities such as landfills and incinerators [13]; and living or working in areas that receive hazardous or illegal waste from improper waste trade [14].

Despite the fact that research began several decades ago on food packaging waste from school foodservice and produced substantive recommendations on how to reduce it, limited success has been achieved in reducing food packaging waste in school foodservice settings. A 1992–1993 study in a Louisiana school district determined that transitioning school foodservice milk packaging from cartons to individual mini-pouches could cut waste volume by 70% and save USD 30,000 annually in operation costs [15,16]; yet milk remains served in cartons in most schools across the U.S. [16,17]. A 2003 study of institutional foodservice (including schools) demonstrated the high expense of comingling organic waste with nonorganic waste in landfill bins, which also resulted in negative environmental impacts [18]. Yet organic waste, or compost, continues to be comingled with landfill-based waste in school settings [19]. In 2010, Prestin and Pearce [20] advocated for consistent and accessible recycling infrastructure, recycling education, and future-oriented value-setting as being essential for improving school waste management, yet such infrastructure, education, and prioritization is rare and inconsistent across schools in the U.S. [19].

Legislation aimed at reducing packaging waste has gained momentum over the last two decades [21]. However, the COVID-19 pandemic disrupted progress by simultaneously causing delays in adopting and implementing new anti-packaging legislation and increasing packaging waste in the food supply chain in the U.S. due to concerns about hygiene [22]. Yet, public policies to reduce waste are starting to be re-visited. In California’s 2021–2022 legislative session, advocacy organizations challenged politicians to implement more restrictions on what is deemed “recyclable” [23], decrease consumer confusion [24], invest in MSW management infrastructure [25,26], and join the 196 countries that have declared the exporting of contaminated plastic scrap to be hazardous and illegal [27]. Maine and Oregon recently passed historic anti-packaging legislation that now requires packaging producers to contribute to covering the operational costs for their products’ disposal [28,29,30].

Although both bottom-up and top-down actions are being taken to reduce MSW, more coordinated research is needed to elucidate which actions are most impactful. Research on school foodservice MSW is particularly important, given it represents at least 1% of overall MSW during the school year [5]. A major obstacle to rigorous and replicable research in this area is the lack of clear and accepted methodologies to evaluate MSW [2]. In 2014, Ward, Wells, and Diyamandoglu [19] called for a standardized evaluation to compare performance on waste reduction and source separation across schools in order to learn from relevant programs.

The objectives of this literature review are two-fold: (1) to describe the methodologies that have been used to evaluate food packaging waste generation and disposal in school foodservice; and (2) to describe and share details of a new practical instrument developed based on the literature reviewed to fill an identified gap in efforts to evaluate food packaging waste generation and disposal in school foodservice.

## 2. Materials and Methods

### 2.1. Search Strategy and Study Selection

The methodology for this systematic review follows the PRISMA (Preferred Reporting Items for Systematic Reviews and Meta-Analyses) guidelines [31] as illustrated in Figure 1.

Peer-reviewed journal articles and grey literature (including master’s theses or doctoral dissertations, and reports conducted by school officials) describing research on food packaging waste, using qualitative and/or quantitative methodologies, were included in our review. Studies conducted in the school (between kindergarten and twelfth grade, excluding specific early childhood education programs) or college/university environments—either the entire school or school foodservice venues—were eligible. Studies conducted in other settings such as hospitals, entertainment facilities, restaurants, and sports stadiums were not included. Study participants included students, administrators, and teachers. Only publications written in English were included. No limitations were placed on the geographical location of the research or the date the research was published.

Two independent researchers searched for relevant studies published before March 2021 using Google Scholar. Following a two-stage search protocol [32,33], upon reviewing the first 500 study titles to determine if the content was related to waste in school settings, researchers identified 61 studies—38 using search terms “solid”, “waste”, “school”, and “cafeteria” and 23 using search terms “food packaging”, “waste”, and “school”. Researchers then identified nine additional studies after reviewing reference sections of the 61 studies identified in the Google Scholar search. Studies not independently selected by both researchers were discussed and evaluated for final inclusion. One study was identified from an email announcement from a peer-reviewed journal that arrived after the original search [34]. This catalyzed a final Google Scholar search at the later date with the same search terms and protocol, but no additional studies were identified. This resulted in 71 studies.

Google Scholar was used for this review as opposed to controlled databases such as PubMed or Scopus because it has a larger database of studies and it includes grey publications [33], which is especially important as this topic is an emerging research field with no singular database. Finally, this research topic does not include clinical information [33], and when search terms were compared across PubMed and Scopus databases, the same studies were found as those in Google Scholar.

Of the 71 relevant studies identified, researchers reviewed each study’s abstract to evaluate study design and methodology, resulting in the elimination of 47 studies. Criteria for elimination included: not conducted in the school or college/university environment or did not focus on school meal program MSW. After screening by abstract, the remaining 35 records were evaluated by their full text. That evaluation resulted in the elimination of 11 studies because they did not focus on foodservice packaging waste, did not have replicable methodology or methodology was not described, or was a curriculum or a practitioner’s guidebook to implementing waste reduction programs rather than a research study. Studies were included in the final review if they reported on three essential variables: clear description of methods used, school type, and environment evaluated. The final review included 24 studies.

### 2.2. Data Extraction

Data were extracted from research studies for the following variables: author(s), year, study design (observational or experimental; pretest–posttest or cross-sectional), methods (e.g., multiple methods, modeling, focus groups, interviews, questionnaires, and waste audits), length of data collection (in days), types (and amounts) of waste categorized, waste bin types used (e.g., landfill, compost, recyclables), country, state/province, school type (elementary school, middle school, high school, college/university), number of schools studied, environment evaluated (whole school or school cafeteria), study participants (e.g., students, staff, and/or administrators), number of study participants, racial and socioeconomic demographics of the school population (rates of free and reduced-price school meals (FRPM), property tax and value, and classification of rural or urban as geographic indicators [35]), intervention, intervention period or data collection period, and publication type (doctoral dissertation, master’s thesis, peer-reviewed publication, or report). These variables were selected based on the prevalence of such data in the studies and the importance of such data to provide context for interpretation of findings.

## 3. Results

The methods varied across the 24 studies reviewed, including objective methods of waste audits, models, and secondary data as well as subjective methods of qualitative observations, questionnaires, interviews, and focus groups. In total, 18 studies used a single approach to assessing waste while 6 studies used multiple methods. Table 1 provides results on some of the data extracted for all reviewed studies. Appendix A provides results on all data extracted for all reviewed studies. Table 2 provides an overview of the different food packaging waste methodologies that have been used in school environments.

### 3.1. Settings, Participants, Designs, and Interventions

Eleven studies occurred at multiple types of schools, including elementary school (kindergarten through fifth grade), middle school (sixth through eighth grade), and high school (ninth through twelfth grade). Five studies were conducted exclusively at elementary schools, two at middle schools, and two at high schools. Four studies focused on college or university campuses. Of the 24 studies that noted their location, 16 were conducted in at least nine different states within the United States and one of the 16 studies occurred across multiple states [36]. The remaining identified locations were Canada, India, Philippines, Romania, and Spain [37,38,39,40,41,42,43].

The number of schools assessed in each study ranged from 1 to 487 with a median of 4.5 (four studies did not identify the number of schools assessed [20,36,43,44]). Most studies (*n* = 14) encompassed the entire school (including administrative offices, classrooms, and restrooms in addition to school foodservice locations), thereby increasing the amount and type of food packaging waste and more general school MSW analyzed [19,20,35,37,38,39,40,41,42,44,45,46,47,48,49], such as silt, soil, and mud [38], textiles [39], or leather [40]. The other 10 studies occurred in school foodservice locations only (e.g., cafeterias) [7,15,18,34,36,43,50,51,52,53].

In both objective and subjective methodologies, there were study participants; in studies with objective methods, study participants generated the waste, while in subjective methods, they participated in the questionnaires, interviews, and focus groups. Students were the sole study participant in 7 of the studies [7,15,20,35,38,50,52]. Staff and administrators were the study participants in 5 studies [18,34,36,48,51]. In the other 12 studies, study participants included all three study participant types: students, staff, and administrators [19,37,39,40,41,42,43,44,45,46,47,49,53].

Ten studies reported socioeconomic data [19,20,34,35,38,45,47,48,52,53] and five studies reported participant race or ethnicity [35,44,45,47,50].

Of the 24 studies included, 14 were observational (no intervention tested) and 10 were experimental (tested an intervention). Twenty studies were cross-sectional and 4 used a quasi-experimental pretest–posttest design [15,19,45,50]. In addition to the 4 pretest–posttest studies, 6 cross-sectional studies evaluated an intervention [44,47,49,51,52,53].

Interventions included implementing a program aimed to reduce waste generation or increase recycling (such as training and informal peer-based education) [15,19,44,51,52], formal education (such as classroom-based or curriculum-oriented programs) [47,50,53], or both [45,49]. Intervention studies involved data collection both before and after the invention was implemented (pretest–posttest) and one-time data collection (cross-sectional). Intervention periods ranged from one day to several weeks [19,50,51,52], one month to one year [44,45,49], or indefinitely [53]; two studies did not report length of intervention [15,47].

**Table 1 ijerph-19-05607-t001:** Characteristics of studies (*n* = 24) included in the systematic review of methods used to evaluate school foodservice packaging waste, sorted by method.

Author(s)	Year	Method(s)	Objective and Subjective Measures	Waste Bin Types	School Type	Environment Assessed	Waste Audit Data Collection Period	Publication Type
Arazo RA [39]	2015	waste audit (sorted + weighed)	food packaging waste categorized: paper and paper products, hard plastics, soft plastics, glasses, metals, woods, food leftovers, yard, textile, inorganic, hazardous, special wastes	trash	college/university	whole school	4-week sampling period	peer-reviewed
Castrejon A [35]	2008	waste audit (sorted + weighed)	food packaging waste categorized: mixed paper, food scrap and yard waste, cans/bottles, trash	trash, recycling (mixed paper), recycling (cans/bottles), compost (food scraps and yard waste)	elementary school	whole school	one 10-week waste audit	master’s thesis
Felder MAJ [42]	2001	waste audit (sorted + weighed)	food packaging waste categorized: animal bedding, animal waste, cloth, food, wood, glass, paper, plastic (#1, 2, 5), plastic (#3, 4, 6, 7), metal (aluminum), metal (ferrous), miscellaneous	trash, recycling, compost (food scraps)	college/university	whole school	three-plus 1-day waste audits per activity area	peer-reviewed
Gallardo A [40]	2016	waste audit (sorted + weighed)	food packaging waste categorized: plastics (PET, HDPE, LDPE, PP, and PS), ferrous metals, nonferrous metals, clean and dirty paper, clean and dirty cardboard, Tetra Brick cartons, glass, organic matter, sanitary cellulose, rubber and leather, toxic and hazardous waste, inert waste	trash	college/university	whole school	two 1-day waste audits	peer-reviewed
Hahn NI [15]	1997	waste audit (sorted + weighed)	food packaging waste categorized: carton, pouch, cardboard, metal cans, compost	trash	K-12	cafeteria	n/a	peer-reviewed
James L [44]	2015	waste audit (sorted + weighed)	food packaging waste categorized: recycling, trash, compost	trash, recycling, compost (food scraps)	elementary school	whole school	three 1-day waste audits	peer-reviewed
Ramamoorthy R [38]	2019	waste audit (sorted + weighed)	food packaging waste categorized: food, paper, silt/soil/mud, plastic, wood/glass/metal/textile, clinical/sanitary, e-waste, other	student waste, campus waste	K-12	whole school	one 5-day waste audit	peer-reviewed
Ravenelle J [52]	2018	waste audit (sorted + weighed)	food packaging waste categorized: trash, recycling, food, liquid	trash, recycling, compost (food scraps and liquid waste)	elementary school	cafeteria	three 1-day waste audits	peer-reviewed
Schumpert K [49]	2012	waste audit (sorted + weighed)	food packaging waste categorized: comingled containers, paper, cardboard, food waste, non-recyclable paper	trash, recycling, compost (food scraps)	K-12	whole school	n/a	report
Schupp CL [53]	2018	waste audit (sorted + weighed)	food packaging waste categorized: reusable food items, compost, recycling, trash	trash, recycling, compost (food scraps)	K-12	cafeteria	one 18-day waste audit	peer-reviewed
Hollingsworth M [51]	1995	waste audit (sorted + weighed + volume)	food packaging waste categorized: food, oil/suet, cardboard, paper, metal, plastic, glass, milk component, plate waste, miscellaneous	trash, recycling	K-8	cafeteria	two 10-day waste audits	peer-reviewed
Baca J [36]	2011	questionnaire	food packaging waste categorized: recycling, trash, compost	trash, recycling, compost	K-12	cafeteria	n/a	master’s thesis
Chan TC [47]	2013	questionnaire	level of MSW practices	n/a	K-12	whole school	n/a	report
Fleckenstein RM [48]	2016	questionnaire	food packaging waste categorized: paper, plastic, Styrofoam, metal	trash, recycling, compost	K-12	whole school	1 day	doctoral dissertation
Iojă CI [37]	2012	questionnaire	level of food packaging knowledge	n/a	K-12	whole school	n/a	peer-reviewed
Vitamog AT [43]	2012	questionnaire	level of MSW practices	n/a	middle school	cafeteria	n/a	peer-reviewed
Wie S [18]	2003	multiple methods—modeling, case studies, interviews	economics of labor, fees, and services	n/a	K-12	cafeteria	n/a	peer-reviewed
Smyth DP [41]	2010	multiple methods—interviews, waste audit (sorted + weighed)	food packaging waste categorized: paper and paperboard, disposable hot beverage cups, beverage containers, plastics, expanded polystyrene, glass, ferrous metals, non-ferrous metals, organic material, hazardous by-products, electronic waste, non-recyclable other; level of MSW practices	trash, recycling, compost (food scraps)	college/university	whole school	two 5-day waste audits	peer-reviewed
Palmer S [34]	2021	multiple methods—interviews, observations	food packaging waste categorized: packaging waste, recycling, food waste; level of MSW practices	trash	K-12	cafeteria	n/a	peer-reviewed
Prescott MP [50]	2019	multiple methods—food systems awareness poster analysis, questionnaire before and after intervention	level of food packaging knowledge	n/a	middle school	cafeteria	n/a	peer-reviewed
Cunningham-Scott CB [45]	2005	multiple methods—curbside recycling reports, control comparison and experiment schools identified, intervention, waste audits (sorted + weighed + got volume) before and after intervention, questionnaire, waste hauler reports	food packaging waste categorized: paper, food waste, cardboard, comingled recyclables, non-recyclable trash; participation rates; level of food packaging knowledge	recycling (mixed paper), recycling (cans/bottles)	elementary school	whole school	5-day waste audit	master’s thesis
Ward MN [19]	2014	multiple methods—case study (tool formation), test case (informal interviews, intervention, waste audit (sorted + weighed) before and after intervention)	food packaging waste categorized: trash, paper, aluminum, plastics; level of MSW practices	trash, recycling (mixed paper), recycling (cans/bottles)	elementary school	whole school	two 5-day waste audits	peer-reviewed
Cagnassola L [7]	2016	modeling	food packaging waste categorized: reusable trays, reusable utensils, plastic utensils, foam trays, compostable utensils, compostable trays	n/a	high school	cafeteria	n/a	master’s thesis
Prestin A [20]	2010	focus groups	level of knowledge, attitudes, and behaviors towards recycling	n/a	high school	whole school	n/a	peer-reviewed

FRPM = free and reduced-price school meals.

**Table 2 ijerph-19-05607-t002:** Methodologies used in the reviewed studies (*n* = 24) to evaluate school foodservice packaging waste.

Method	Description	Outcome	Components	Author(s)	
objective—waste audit	objective measurement of type and amount of waste generated	mass and/or volume of waste generated by waste type(s)	landfill, recycling, compost, and/or other more granular categories	Arazo RA [39] Castrejon A [35] Cunningham-Scott CB [45] Felder MAJ [42] Gallardo A [40] Hahn NI [15] Hollingsworth M [51]	James L [44] Ramamoorthy R [38] Ravenelle J [52] Schumpert K [49] Schupp CL [53] Smyth DP [41] Ward MN [19]
objective—secondary data	external, pre-compiled data	holistic perspective on direct and indirect influences of waste generation and disposal	waste hauler reports; curbside recycling participation; socioeconomic information	Cunningham-Scott CB [45] Prescott MP [50] Ward MN [19] Wie S [18]	
objective—model	compile data to project conditions in the near- or far-term	mass and/or volume of waste generated cost(s) of disposable food packaging, operations, and/or food packaging waste	landfill, recycling, compost, and/or other more granular categories;monetary costs	Cagnassola L [7] Cunningham-Scott CB [45] Wie S [18]	
subjective—observations	observe (real-time or through photographs) meal prep and serving operations as well as disposal practices for their associated food packaging waste types and amounts	when different types of food packaging waste are generated and how they are disposed of;capture and demonstrate practices and interventions	kitchen/prep, serving, disposal; photographs taken; no-blind, blind, or double-blind analysis	Palmer S [34] Prescott MP [50]	
subjective—questionnaires, interviews, or focus group	study participants’ perceptions of barriers to/facilitators of reducing waste; collected individually or in a group setting	data from many people on specific topics (e.g., barriers and facilitators to waste reduction; knowledge, behaviors, and attitudes of waste reduction practices)	knowledge; behavior; attitude	Baca J [36] Chan TC [47] Cunningham-Scott CB [45] Fleckenstein RM [48] Iojă CI [37] Palmer S [34] Prescott MP [50] Prestin A [20] Smyth DP [41] Vitamog AT [43] Ward MN [19] Wie S [18]	

### 3.2. Methodologies

As indicated in Table 2, the methods employed by the 24 studies included primary data collection (both objective and subjective data), use of secondary data, and modeling based on primary and/or secondary data. We describe each of these methods in more detail below.

#### 3.2.1. Waste Audits (Objective Data)

The dominant method related to MSW evaluation was a waste audit (*n* = 14), the act of quantifying various categories of MSW via objective measures of weight and/or volume. Most studies using waste audits conducted a one-time waste audit (*n* = 12), while two conducted a waste audit before and after an intervention [19,45]. Waste audit data collection periods ranged from two 1-day audits [40] to one 4-week audit [39].

All waste audits included in the review consisted of the same general structure: the collection of waste from waste bins, sorting the collected waste into categories, then capturing the weight and/or volume of the different waste categories. Bin collections were limited to the study setting (e.g., cafeterias or whole school) or randomly selected if the sheer volume of potential waste collection was too high [41]. Most waste audit studies (*n* = 12) had a “trash” bin for non-recyclable or non-compostable material [15,19,35,39,40,41,42,44,49,51,52,53]. Ten studies had a “recycling” bin for material considered recyclable for their specific location [19,35,41,42,44,45,49,51,52,53]. Seven studies had “compost” bins for food waste and other organic materials [35,41,42,44,49,52]. The study settings that had multiple types of waste bins were able to capture more detailed data on waste generated during the waste audit [38,39,40,45,51,53].

Once the waste was collected from the waste bins, researchers sorted that waste into waste categories for analysis. The intention behind the sorted waste categorization differed between studies. Three examples of different intentions resulting in different sorted waste categories include regionally defined categorizations [41], a green schools certification [44], and self-defined categories based on the school’s needs and interests [42].

Two sorted waste categories included in all waste audits (*n* = 14) were recycling (e.g., mixed paper, cans/bottles) and trash (e.g., inert waste, non-recyclable trash). All but one study [19] also categorized compost (e.g., food waste, organic material) as a type of waste. One study categorized “rescuable food” (e.g., whole apple) and “miscellaneous packaged foods” (e.g., packaged crackers) [53] and another study categorized liquid waste (e.g., partially consumed milk) [52]. Mixed paper and cardboard were categorized in two studies: in one study they were lumped together [35] and in one study they were separated [40]. In three studies, metal was a catch-all category [38,39,51], while in five studies there were more finite categorizations of metals, such as ferrous versus non-ferrous metals [15,19,40,41,42]. Napkins were a unique category in only one study [34] and were included in the “clean and dirty paper” category in another study [40]. Plastic was a catch-all category for all plastic types in four studies [19,38,41,51]. Several studies included plastic subcategories: hard plastic and soft plastic [39] and categorization of plastics by their resin type or resin code (the number inside the chasing arrows recycling symbol) [40,42]. None categorized plastics by their function (e.g., utensils, clamshells). Reusable foodware—which is not waste but instead foodware that can be reused after being cleaned—was only categorized in one study [7].

All studies measured sorted material by capturing weight data by the sorted categorization. Two studies also included volume data [45,51]. One of the studies that included volume data [51] did so after implementing an intervention of changing milk packaging from cartons to plastic pouches. As expected, both weight and volume of milk packaging decreased from the intervention, but the magnitude of the reductions differed: weight decreased by 23%, volume decreased by 167%. The other study [45] evaluated the impact of measuring material volume as it is typically tossed into a waste bin versus one that is stacked like how researchers place sorted waste into categories. Study participants in one school emptied and stacked their lunch trays resulting in about half the waste volume as a school that did not have participants stack their lunch trays.

With regards to weight data, the categorical amounts varied by when the data were collected: in one study [44] the pre-sort weight of material in the waste bins was 63% landfill, 32% recycling, and 5% compost. However, once sorted, only 7% of the discarded material was landfill, while 67% of it was recyclable and 26% compostable.

#### 3.2.2. Secondary Data (Objective Data)

Secondary data included waste hauler reports (e.g., quantity, type, and frequency of waste collected by the waste service provider) [45] and curbside recycling participation (e.g., how the number of residents participating in curbside recycling programs relates to recycling practices in the school) [45].

#### 3.2.3. Modeling (Objective Data)

Two studies leveraged modeling software [7,18], where they gathered data from multiple sources (e.g., weights of alternative food packaging materials, costs of alternative operations) to predict long-term outcomes (e.g., amount and type of food packaging waste, costs) based on specified conditions.

#### 3.2.4. Subjective Data

Subjective data was collected via qualitative observations, questionnaires, interviews, and focus groups. Observations were of the amount and types of food packaging waste in meal prep within the kitchen via structured qualitative observation [34] as well as student consumption and sorting behavior within the cafeteria via photographs [50]. Questionnaires aimed to capture data on food packaging knowledge, attitudes, and behaviors [36,37,43,45,47,48,50] from students [50]; staff and administrators [36,48]; or students, staff, and administrators [37,43,45,47]. Four studies had interviews [18,19,34,41] and one study had a focus group [20], which were also focused on the knowledge, attitudes, and behaviors of food packaging waste in foodservice.

## 4. Discussion

The objective of this systemic literature review was to understand which methodologies exist to evaluate food packaging waste in school environments to inform MSW and school foodservice professionals and researchers as well as describe and share a novel waste audit tool developed and field tested. While literature was sourced from across the world, because the majority of studies identified were conducted in the United States, the discussion is most relevant to the U.S. context.

### 4.1. Settings and Participants

There did not appear to be an influence in results based on the number of schools analyzed. Studies that evaluated waste collected throughout the school environment versus specifically from a food-only location (e.g., cafeteria) had far more variation in waste categories, particularly with non-food packaging wastes. Narrowing data collection to locations at the school site where school foodservice packaging waste is most typically disposed is recommended. This can create less contamination from MSW other than that of food packaging waste generated by school foodservices, and thus more precision for waste sorting and the waste categories’ volumes or weights.

Presently, school foodservice waste assessments do not routinely or comprehensively analyze racial or socioeconomic data with regards to the gravity of the waste generation problem and opportunities for advancing waste reduction. As noted in the introduction, economically disadvantaged communities and communities of color disproportionately experience negative health impacts of MSW as well as are more likely to participate in school foodservice programs. It is therefore important to better understand if there are compounded human health impacts by also consuming food from school foodservice programs with disposable packaging. Furthermore, since school foodservice packaging is such a large contributor to MSW in the U.S. and it is becoming clearer that certain types of packaging have greater human and environmental harm [8,9], overlaying the packaging types with harms and demographics could guide the prioritization and avenues of resolutions. Recommendations for racial data to collect include more racial categories such as Asian or Pacific Islander, Black or African America, Hispanic or Latino, Native American or Alaskan Native, White or Caucasian, Multiracial or Biracial, or a race/ethnicity not listed here [54,55,56]. Recommendations for socioeconomic data to collect include FRPM rates or accurate proxies, such as property value or parental income, education, and occupation [57]. This would be especially important to analyze with regards to access to and participation in school foodservice programs [58]. While we did not include such metrics in our WASTE instrument, demographic data will be captured in the study which was the impetus for the instrument’s development.

### 4.2. Methodologies and Their Areas for Improvement

This literature review gave important insights for optimizing a waste audit protocol for school foodservice. Waste audits start with the collection of waste from study participants via waste bins. More detailed waste information was derived from studies with multiple waste bin types, likely due to less contamination within a waste bin, allowing the researchers to better sort and discern material [59]. Given the inconsistency in waste bin types, a standardized waste audit instrument should include all waste bin types allowing for variability between study sites.

There was no uniformity on the post-sort waste categories used for evaluation or the number of days data were collected, as they were determined by regional guidance, certification process, or the prioritization for actionable insights for the studied school(s). The number of days required to accurately assess school foodservice packaging waste will depend on how variable the packaging is from day to day. Some school foodservice waste, such as utensils and milk cartons, may be used every day, while others may be specific to the menu item. Many schools utilize cycle menus wherein items are repeated periodically (e.g., every 2–4 weeks). School menus should be reviewed with school foodservice staff to understand variability in the types of packaging used to inform the number of days waste assessments are needed. Furthermore, the difference in data collection periods is often due to time or resource restrictions, methodology, requirement to capture a certain amount of data for evaluation, and if there is seasonality of waste generation. Standardization of post-sort waste categorization, especially for some of the harder to manage materials (e.g., plastics) or more prevalent materials (e.g., compost) could elucidate opportunities for programmatic or educational action as well as comparison across interventions and intervention periods.

One of the other more inconsistent components of the examined waste audit protocols was collecting weight and/or volume data on waste. This type of data capture is often precipitated by the practical application of research learnings; weight data are used as a general metric within the MSW industry while volume data determine a waste service provider contract; thereby determine the cost for such services. So, if learnings are based on weight, it is harder to apply them in terms of the volume of carts and frequency of collection while if learnings are based on volume, then it is harder to apply them to the larger waste trend data.

An additional complication is when to collect weight and/or volume data. Capturing such data only after the researcher sorted material into categories provides limited learnings, such as no insight into the accuracy at which study participants sort their waste into the proper bin. However, volume data can vary substantially between pre-sort and post-sort due to the differences in study participants tossing the waste into bins and data collectors neatly stacking the waste by categories. That waste-stacking practice does not necessarily reflect real-life waste practices. Therefore, if MSW volume is of interest to the research team, it is important to determine when and how volume data will be collected.

Since there are important applications to both volume and weight data, we capture both types of data in the instrument we developed. Furthermore, those period-in-time data captures are ideally standardized during the data collection process, such as before the sorting occurs or after all material is correctly identified and categorized. A recent development in waste assessment is capturing data on reusable foodware. The intent behind this is to give insight into preventative measures taken to address foodware generation and disposal issues and the behavioral components coupled with the use of reusable foodware. Furthermore, it gives greater insight into the entire ecosystem of consumption within this space, to better identify barriers and opportunities for improvement to achieve desired goals on waste mitigation.

The second grouping of methodologies reviewed was questionnaires. Like all other components of food service waste examined, questionnaires could benefit from the standardization of intent, length, and audience. These methodologies have limited use for objectively measuring amounts of waste and therefore are not included in our developed instrument. However, they do serve a purpose: in addition to collecting sociodemographic and other contextual data, questionnaires can provide insights into types of waste, waste reduction strategies that are or have been used, and barriers and challenges with handling waste.

The studies reviewed with multiple methods gave relatively comprehensive insight into the larger food service waste generation system in schools. A multiple methods approach of combing waste audits with mealtime observations can provide greater insight into the behavior before and during packaging waste generation and disposal, or that decisions and actions taken by stakeholders before or outside of the actual disposal of the MSW are crucial to decreasing the amount of MSW.

The inclusion of various data sources aids comprehensive insight. For instance, waste hauler reports may provide an additional layer of data quality assurance or more perspective on the larger MSW system, such as potential educational or infrastructural spillover effects where waste reduction programs in schools coincide with higher recycling rates in their study body’s residential jurisdiction(s). Or modeling software could be used to conduct analyses by triangulating MSW data for scaled projections, both in space and time, to demonstrate potential MSW generation trends. Furthermore, photovoice—the use of ethical photography by participants to capture and demonstrate practices and interventions—is increasingly used as a participatory research approach to deepen engagement with the environment of analysis, providing additional perspective on the system of study [60]. We recommend the use of waste hauler reports, modeling software, and/or photovoice to incorporate a more participatory method to MSW analysis as well as glean deeper qualitative insights pertaining to MSW behavior and attitudes. Additionally, the U.S. EPA’s WARM model [61] would be a worthwhile secondary analysis for researchers looking to quantify greenhouse gas impact after initial waste audit data are collected. These additional data sources can influence more comprehensive food packaging waste reduction and management policies like in the European Union [62,63].

There are a couple of notable limitations to this systematic literature review. First, research on food packaging waste in school foodservice programs remains limited, thereby not providing a large corpus of studies to analyze. Second, we only analyzed studies in English, which likely further reduced the pool of applicable studies. Third, 16 of the 24 studies were conducted in the U.S., likely influencing the types of conclusions drawn. Finally, we were not able to compare the food packaging waste results in school foodservice programs across the reviewed studies because of the lack of standardization in methodology, even within a specific method (e.g., waste audits).

Considering the preceding summary, we propose a new instrument to analyze food packaging waste in foodservice in the school environment.

### 4.3. New Instrument Creation and Refinement

The research team created a new instrument informed by the literature review—Waste Audit for Sustainable Transitions and Evaluations (WASTE)—for quantifying and categorizing food packaging waste generated and disposal methods based on data needs to be used alone or in conjunction with plate waste data collection methods [64]. The WASTE instrument—found in the Appendix A—was informed by results from the literature review and preexisting practitioner instruments often used in non-school settings, such as the U.S. Green Building Council’s TRUE Zero Waste certification [65], the Post-Landfill Action Network’s Waste Audit Manual [66], and the University of California Global Food Initiative’s Waste Auditing Practices [67]. Some specific components of WASTE developed during that process included determining needed data and use(s) for the data prior to collecting them [36], school-specific nuances such as administrative participation in waste reduction efforts and the seasonality of school waste generation [37], and capturing the waste bin location [38]. An additional preexisting practitioner instrument is the U.S. EPA Waste Reduction Model (WARM) [61], which is similar to a waste audit; however, it focuses more on the greenhouse gas impact of the data collected versus how to collect the data. The WARM model helped inform the necessary categorization of food packaging waste materials (e.g., compostable plastic and compostable paper) in WASTE.

After the initial creation of WASTE, we field-tested the instrument in conjunction with plate waste data collection at one elementary school in February 2020. Field testing results informed updates to WASTE to include meal type (e.g., breakfast, lunch, snack, supper), the collection of a specimen per foodware type (e.g., utensil, plate), clarify material types (e.g., capturing a plastics resin code, which is the number stamped inside the chasing arrows symbol), and add a notes section to describe environmental components such as waste bin signage or abnormalities.

The systematic review of food packaging waste in school foodservice elucidated the need to include four additional components in our developed instrument: weight and volume, where the waste data are collected, photographs of signage, and not collecting data on reusable foodware.

The largest finding in the systematic review to inform our instrument was the methodological difference in waste audits of collecting weight and/or volume data and when to do that collection. Weight data can inform different pathways towards waste reduction compared to volume data, especially given the context and goals of the study. We therefore updated the WASTE instrument to capture both types of data so that researchers can determine which (or both) is best for their study. As for when to collect data, categorical weight and/or volume waste amounts can change pre- and post-sort, so our instrument includes both options. Our instrument also allows and specifies if volume data are captured through neatly stacked versus tossed approaches.

The second largest refinement to our instrument was including where the waste was generated and thus collected. A distinction that was not well explored in the literature is school foodservice waste generated during the process of food preparation (e.g., in the ‘back of the house’) versus waste generated by students in the process of food service (e.g., in the ‘front of the house’). Because both may be of interest, we designed our instrument to have the flexibility to assess both.

The third modification to our instrument was photography. While our original instrument included photographs of a typical meal (Figure 2), the use of photography in one study [50] spurred us to include photographs of waste bin receptacles and waste bin signage. The use of photography in waste evaluations is new and provides important consideration to not only allow researchers to review material after data collection and thus gain additional, independent validation of information documented on the instrument, but also could demonstrate a connection(s) between the larger MSW ecosystem, such as educational signage and how material is sorted into the available waste bins. We recommend a qualitative analysis strategy for the photographs like Prescott et al. [50], which included two, trained researchers independently evaluating the photographs based on the research questions (e.g., plate waste amounts [50], or school foodservice operations, infrastructure, and packaging waste types and quantities).

The final update to the WASTE instrument was omitting data capture on reusable foodware. Our instrument does not include any categorization for reusable foodware because the instrument is strictly for waste audits and reusable foodware would not be part of the waste stream. The impacts of reusable foodware on food packaging waste generation, however, could be evaluated if multiple waste audits occurred pre and post implementation of a reusable foodware intervention. Otherwise, the reviewed literature validated the items and material types already outlined in the instrument, which we kept rudimentary to allow for application across highly variable food packaging waste types in school foodservices.

Of note: while the waste audit instrument is focused specifically on waste audits, the systematic review substantiated the need for additional data and complementary methods, such as modeling, waste hauler reports, and socioeconomic information. We recommend all waste audits include such additional data as study resources allow.

## 5. Conclusions

In studies aimed at understanding the quantity and type of food packaging waste in school foodservice programs, there is substantial variation in the settings, participants, designs, interventions, and methodologies. While each study to date has generated valuable insight into the specific context of which the study occurred, the variation in data collection and analysis has stunted the growth of the research field at large through seemingly ungeneralizable results [34].

Informed by this literature review, we created a new, credible food packaging waste audit instrument to standardize the collection and analysis of packaging waste in school foodservice programs in the U.S. Future research on this novel methodology will be conducted to ensure it is both feasible for use and reliable in multiple school foodservice environments. The creation of a standardized methodology is imperative [40], especially since the most recent “novel design of a solid waste audit” was created in 1998 [42] and the only standardized items to date pertain to program implementation [19] versus data collection and analysis. A standardized tool can help future researchers and practitioners gather baseline data, understand the impact of MSW generated with regards to any intervention tested, learn from similar initiatives, and develop strategies to achieve waste goals. This standardization will be increasingly important as food packaging trends change (e.g., movement to reusable foodware and the adoption of compostable or biodegradable material) and more research to assess school foodservice food packaging waste is conducted in the U.S. to minimize environmental and human health harms.

## Figures and Tables

**Figure 1 ijerph-19-05607-f001:**
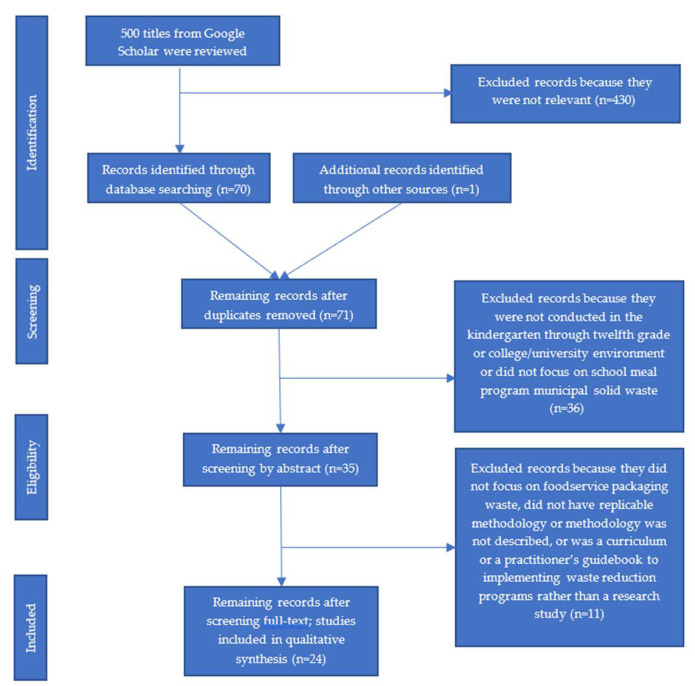
The flow of studies through the different phases of the systematic review.

**Figure 2 ijerph-19-05607-f002:**
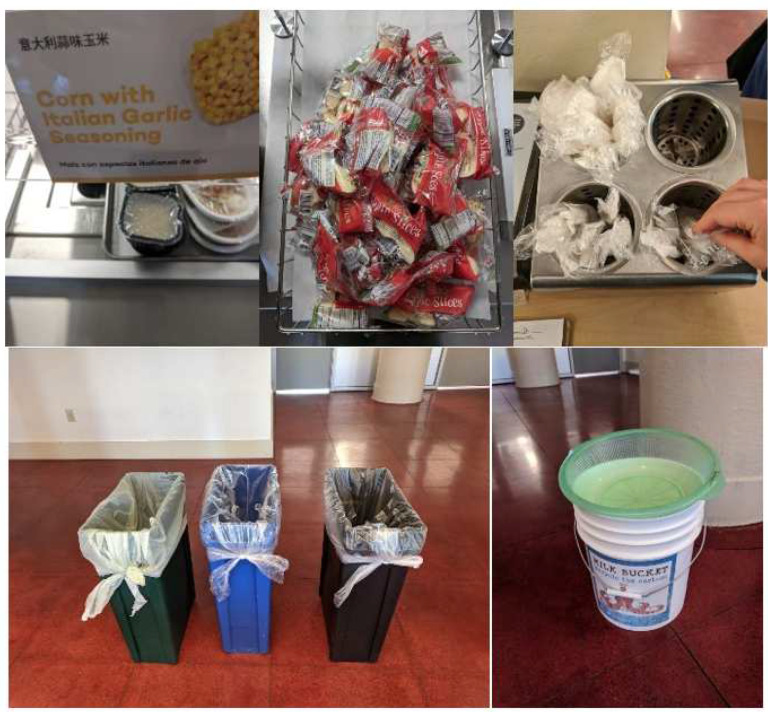
Photographs of a typical meal captured while field testing the newly developed WASTE instrument.

## Data Availability

Not applicable.

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
