# Peer review of "Evaluating Food Packaging Waste in Schools: A Systematic Literature Review"

_ijerph, 2022, doi:10.3390/ijerph19095607_

Round 1
Reviewer 1 Report
This research evaluates the different methodologies for quantifying packaging waste. It is based on different investigations previously carried out and in the end it proposes a new methodology that is documented in the supplementary material. Due to the tendency to take care of the world, in the sense of circular economy and/or food loss and waste, I believe that this review is very valuable because it provides information that will serve as a school guide for the quantification of waste and in this way, take the necessary actions to reduce it. Despite being a review, they describe the methodology, but I think it is justifiable because they clarify that it was a systematic review and the criteria used to carry it out. That is why this manuscript must be accepted with minor corrections:
Please review the cited literature, since some of the cites do not have DOI, and it is important to add it.

Author Response
This research evaluates the different methodologies for quantifying packaging waste. It is based on different investigations previously carried out and in the end it proposes a new methodology that is documented in the supplementary material. Due to the tendency to take care of the world, in the sense of circular economy and/or food loss and waste, I believe that this review is very valuable because it provides information that will serve as a school guide for the quantification of waste and in this way, take the necessary actions to reduce it. Despite being a review, they describe the methodology, but I think it is justifiable because they clarify that it was a systematic review and the criteria used to carry it out. That is why this manuscript must be accepted with minor corrections.
Please review the cited literature, since some of the cites do not have DOI, and it is important to add it.
Response: We have added DOIs where applicable (L570-708).
Line 57-59 I think it's unnecessary to mention communities of color. Mentioning the impact on low-income communities is enough. Or why should be mentioned?
Response: We feel it is important to mention communities of color as those are not synonymous with low-income communities and communities of color can experience separate and diverse impacts from the food packaging supply chain compared to all-white communities, regardless of their income.
Line: 104 It is not necessary to put a subtitle of "objectives" mentioning them is enough.
Response: We removed “objectives” from L105 and also removed “rationale” from L38 for consistency and to streamline subheadings under the Introduction.
Materials and Methods I have no comments because its description is good and justified
Results: I think that according to the mentioned objectives and the type of study, the results are pertinent. Discussion: I think the discussion section, according to the type of investigation reported, is relevant and good.
Response: Thank you for these comments.
Conclusion section. Line 500-507 The conclusions are very long, please be more concise and clearer, remove unnecessary text (Line 500 to 507).
Response: We removed L524-531 of the conclusion.
Reviewer 2 Report
Dear Authors,
The prepared manuscript concerns a very important issue which is waste. Education in this direction is very much needed.
The manuscript is well organized and prepared. The topic of the work is justified and fully described. It can be used to educate young people at the school stage in order to properly manage waste, of which we currently have a lot.
The manuscript is based on the literature from the last 10 years in 99%.
There are minor editorial flaws that need to be corrected.
The manuscript is most suitable for publication in Int. J. Environ. Res. Public Health.
Author Response
Dear Authors,
The prepared manuscript concerns a very important issue which is waste. Education in this direction is very much needed.
The manuscript is well organized and prepared. The topic of the work is justified and fully described. It can be used to educate young people at the school stage in order to properly manage waste, of which we currently have a lot.
Response: Thank you for these comments.
The manuscript is based on the literature from the last 10 years in 99%.
There are minor editorial flaws that need to be corrected.
The manuscript is most suitable for publication in Int. J. Environ. Res. Public Health.
Response: Thank you for your comments. We made editorial changes based on this feedback and the suggestions from the other reviewers (L119-159, 325-327, 524-531, 540-552, 570-708).
Reviewer 3 Report
The objective of this manuscript is clear and also very interesting. However I have some considerations for authors.
From line 122 to 164 I proposed to synthesize. It is quite long.
In particular, in my opinion I would reduce the paragraph three that is too long. I would suggest summarizing the part of the discussion as well. It would improve the readability of the work.
If I understood correctly, the new recommended tool speaks of photographs to explain a phenomenon. The authors could consider explaining what is the research method (qualitative analysis I guess) to use in order to evaluate the photos collected in a scientific analysis.
Author Response
The objective of this manuscript is clear and also very interesting.
Response: Thank you for these comments.
However I have some considerations for authors.
From line 122 to 164 I proposed to synthesize. It is quite long.
In particular, in my opinion I would reduce the paragraph three that is too long. I would suggest summarizing the part of the discussion as well. It would improve the readability of the work.
Response: We made numerous edits (L119-159) to make our search strategy and study selection more concise. We do not mention our methodology in the Discussion section, so we were unclear what actions to take there.
If I understood correctly, the new recommended tool speaks of photographs to explain a phenomenon. The authors could consider explaining what is the research method (qualitative analysis I guess) to use in order to evaluate the photos collected in a scientific analysis.
Response: We added L499-503 to provide more context into a qualitative analysis of the photographs taken in conjunction with the WASTE instrument.
Reviewer 4 Report
The paper is addressing a very important and timely topic. The paper is well written; however, the following suggestion/recommendations may help to clarify the findings.
- Since authors stated that there is a large variation on the methodology of each reviewed paper exists. Can authors separate the studies and compare the results of the same data collection format (example: waste audit vs. subjective evaluation)?
- There are a few studies (n=24) reviewed. Can we draw any conclusion?
- The proposed WASTE instrument was filed tested in a small setting (elementary school). Can this instrument applicable and practical in larger and different setting (restaurants, household)?
- The WASTE instrument is a good approach but there is some limitation as it is a large variation among the data collection methods as mentioned by authors as well.
The waste quantification is very challenging and unless there are standard and universal method of data collection is in place these results can be close to estimation. The researchers and food packaging industries are looking more and more to develop a biodegradable packaging approach.
There have been challenges in quantifying the food waste as well as food packaging waste. There are some progresses in reducing the FPW waste by avoiding Styrofoam and plastic containers that are considered harmful for the environment. Hence more research is needed in biodegradable material (plant-based) with low greenhouse gas emission for food container. In addition, the following could ben helpful• Abstract is not clearly explained the actual results of this review study. Add some specific data to support the findings. • Are there any good suggestions? • I tried to understand the focus of the paper. It would be helpful and effective if the emphasis has been on the instrument that is proposed to support comprehensive and replicable data. Also, the importance of this study and environmental impact.
Author Response
The paper is addressing a very important and timely topic. The paper is well written; however, the following suggestion/recommendations may help to clarify the findings.
Since authors stated that there is a large variation on the methodology of each reviewed paper exists. Can authors separate the studies and compare the results of the same data collection format (example: waste audit vs. subjective evaluation)?
Response: We compared the methods used in studies based on the criteria of year, method(s), objectives and subjective measures, waste bin types, school type, environment assessed, waste audit data collection period, and publication type (Table 1, p.6-10). We did not compare the results of any of the studies because the focus of the review was on methodology, especially given how disparate the data collection methods were, even within the same type of methodology (e.g., waste audit). In the Discussion, we recommend that future research compare results across studies using the standardized methodology proposed through the WASTE instrument. We also included a table (Table 2, p.10-11) that buckets the methodologies and provides a description, outcome, and components for those methodologies. We deemed that results by methodology were not currently feasible based on the lack of standardization within each methodology. We have added this to the limitations section (L441-445).
There are a few studies (n=24) reviewed. Can we draw any conclusion?
Response: The limited number of studies was due to both the emerging nature of the field (L159) as well as the parameters put in place to select studies for analysis (section 2.1).
The proposed WASTE instrument was field tested in a small setting (elementary school). Can this instrument applicable and practical in larger and different setting (restaurants, household)?
Response: We have not field tested the WASTE instrument outside of the school setting. We hope to include that in a future manuscript once more analysis is done with the WASTE instrument: that future studies test the applicability of the instrument in non-school settings such as restaurants and households, or even hospitals, concert halls, and transit hubs.
The WASTE instrument is a good approach but there is some limitation as it is a large variation among the data collection methods as mentioned by authors as well.
The waste quantification is very challenging and unless there are standard and universal method of data collection is in place these results can be close to estimation. The researchers and food packaging industries are looking more and more to develop a biodegradable packaging approach. There have been challenges in quantifying the food waste as well as food packaging waste. There are some progresses in reducing the FPW waste by avoiding Styrofoam and plastic containers that are considered harmful for the environment. Hence more research is needed in biodegradable material (plant-based) with low greenhouse gas emission for food container. In addition, the following could be helpful
Response: Thank you for your comments. We agree. With this material trend in mind, the WASTE instrument was designed to capture and categorize biodegradable (compostable) material. We added a point to that trend in the Conclusion (L549-550).
- Abstract is not clearly explained the actual results of this review study. Add some specific data to support the findings.
Response: This manuscript focused on the methodologies within the literature, not the results of the studies, so the Abstract entails data garnered from that analysis of methodologies: objective and subjective methods, large variation in identified criteria, that a new instrument was developed, and that such replicability through a standardized instrument can result in further understanding of school foodservice food packaging waste in the U.S. Analysis of results of the literature reviewed was not possible given the disparities in methods used.
Response continued: To the Abstract, we added the number of reviewed studies conducted in the U.S. (L26), that we focus on the U.S. with the new instrument (L33), and the lack of standardization in the methodology (L30-31). We also added both of those to the limitations section (L441-445).
- Are there any good suggestions?
Response: We have suggested the use of the newly created WASTE instrument to standardize school foodservice food packaging waste methodology (L542-552). We feel any other suggestions (e.g., policy) are out of scope of this manuscript.
- I tried to understand the focus of the paper. It would be helpful and effective if the emphasis has been on the instrument that is proposed to support comprehensive and replicable data. Also, the importance of this study and environmental impact.
Response: The first of the two objectives is on the creation of the WASTE instrument (L106-108). The importance of the study is the justification for the creation of the WASTE instrument, which is described in the final paragraph of the manuscript (L538-552). We felt discussing the environmental impact was out of scope since this manuscript focuses more on methodology than on results.
Reviewer 5 Report
The submitted manuscript entitled “Evaluating food packaging waste in schools: a systematic literature review” concentrates on the important issue of evaluating food packaging waste generation and disposal in school foodservice. The final literature review included 24 studies conducted in school environments, 2/3 of which were in the United States. The remaining identified locations were Canada, India, Philippines, Romania, and Spain.
Throughout the whole manuscript references are made almost entirely to the situation in the United States. In very few parts - for example L46-48 - some data from another region are shown. BTW please check again the data from the 2018 World Bank Group report - source [1] – data from 2016.
The Abstract, Introduction and Conclusions lack any kind of information on MSW, waste in schools or anti-packaging legislation in European countries, Canada or Asia. I would strongly recommend looking up and adding such information (instead of the cases of individual states) in order to give the Reader a more global overview of the situation (as the Title suggests). COVID-19 did not disrupt progress in adopting and implementing new anti-packaging legislation increasing packaging waste in the food supply chain in the EU (L85-87) – if this is, according to the Authors, the case in the US please make sure you make it clear to the international Reader.
Some examples of important non-US sources:
Waste Framework Directive (europa.eu)
00c0b3fe-db08-4076-b39a-e92015ce99e0 (europa.eu)
Municipal waste management across European countries — European Environment Agency (europa.eu)
If this is not done, then a modification of the Title is needed or an explanation in the Abstract should be given that Authors concentrate in their discussion on the US context which differs from other regions, incl. the EU (where the circular economy, waste management etc. are high on the political agendas).
Author Response
The submitted manuscript entitled “Evaluating food packaging waste in schools: a systematic literature review” concentrates on the important issue of evaluating food packaging waste generation and disposal in school foodservice. The final literature review included 24 studies conducted in school environments, 2/3 of which were in the United States. The remaining identified locations were Canada, India, Philippines, Romania, and Spain.
Throughout the whole manuscript references are made almost entirely to the situation in the United States. In very few parts - for example L46-48 - some data from another region are shown. BTW please check again the data from the 2018 World Bank Group report - source [1] – data from 2016.
Response: We reviewed data from the 2018 World Bank Group report – source [1] – and on page 3 they state: “global waste is expected to grow to 3.4 billion tonnes by 2050”. Tonnes are metric tons. We have therefore revised L42-43 from “increase to 3.5 billion metric tons” to “increase to almost 3.5 billion metric tons”.
The Abstract, Introduction and Conclusions lack any kind of information on MSW, waste in schools or anti-packaging legislation in European countries, Canada or Asia. I would strongly recommend looking up and adding such information (instead of the cases of individual states) in order to give the Reader a more global overview of the situation (as the Title suggests).
Response: We have made updates to the manuscript to make it clearer that while the reviewed literature is sourced from around the world, the Discussion and Conclusion focus on the U.S. context.
COVID-19 did not disrupt progress in adopting and implementing new anti-packaging legislation increasing packaging waste in the food supply chain in the EU (L85-87) – if this is, according to the Authors, the case in the US please make sure you make it clear to the international Reader.
Response: We updated this reference to COVID-19 (L88) to make it clear that these data pertained to the U.S.
Some examples of important non-US sources:
Waste Framework Directive (europa.eu)
00c0b3fe-db08-4076-b39a-e92015ce99e0 (europa.eu)
Municipal waste management across European countries — European Environment Agency (europa.eu)
If this is not done, then a modification of the Title is needed or an explanation in the Abstract should be given that Authors concentrate in their discussion on the US context which differs from other regions, incl. the EU (where the circular economy, waste management etc. are high on the political agendas).
Response: We have updated the Abstract (L33) to note that the Discussion and Conclusion are focused in the U.S. We have also updated the Discussion (L325-327, 441-443) and Conclusion (L540, 551) to underscore its focus on the U.S.